# Tea Compounds and the Gut Microbiome: Findings from Trials and Mechanistic Studies

**DOI:** 10.3390/nu11102364

**Published:** 2019-10-03

**Authors:** Timothy Bond, Emma Derbyshire

**Affiliations:** 1Tea Advisory Panel, 71-75 Shelton Street, Covent Garden, London, WC2H 9JQ, UK; tim@teaandherbalsolutions.com; 2Nutrition Insight Ltd, Surrey KT17 2AA, UK

**Keywords:** tea, tea compounds, gut microbiome, polyphenols

## Abstract

In recent years, the gut microbiome has become a focal point of interest with growing recognition that a well-balanced gut microbiota composition is highly relevant to an individual’s health status and well-being. Its profile can be modulated by a number of dietary factors, although few publications have focused on the effects of what we drink. The present review performed a systematic review of trials and mechanistic studies examining the effects of tea consumption, its associated compounds and their effects on the gut microbiome. Registered articles were searched up to 10th September 2019, in the PubMed and Cochrane library databases along with references of original articles. Human trials were graded using the Jadad scale to assess quality. Altogether 24 publications were included in the main review—six were human trials and 18 mechanistic studies. Of these, the largest body of evidence related to green tea with up to 1000 mL daily (4–5 cups) reported to increase proportions of *Bifidobacterium*. Mechanistic studies also show promise suggesting that black, oolong, Pu-erh and Fuzhuan teas (microbially fermented ‘dark tea’) can modulate microbial diversity and the ratio of *Firmicutes* to *Bacteroidetes*. These findings appear to support the hypothesis that tea ingestion could favourably regulate the profile of the gut microbiome and help to offset dysbiosis triggered by obesity or high-fat diets. Further well-designed human trials are now required to build on provisional findings.

## 1. Introduction

Tea has been drunk for thousands of years as part of a regular daily habit by people of all ages and is the most frequently consumed beverage globally alongside water [1,2]. Evidence in relation to the health benefits of tea is mounting, with encouraging data implying roles in metabolic and cardiovascular health [3]. Alongside this, the evidence-base around tea ingestion, its associated compounds and aspects of gut health has also been progressively evolving.

The gut microbiome has received considerable scientific attention over the last decades [4]. We know that trillions of microbes have evolved and reside on and within human beings—with cross-talk between microbes and the host becoming increasingly apparent [5]. The gut microbiota had been likened to a new body organ with the analogy that it is an ‘immune system’; comprised of a collection of cells (microbes) that can work in unison with the host and promote health but, equally, initiate disease if it goes off kilter [6]. Altered gut bacterial composition (dysbiosis) is one factor now known to be involved in the aetiology of inflammatory diseases and infections [7]. Indeed, nowadays, it is recognised that a well-balanced gut microbiota composition is highly relevant to an individual’s health status and well-being, helping to prevent and manage chronic diseases [8,9].

Many factors can modulate the profile of gut microbiota. This includes genetic and environmental factors along with baseline intestinal and generic health [10]. The diversity and composition of gut microbes is dynamic with antibiotic usage, obesity, allergies, inflammatory diseases and metabolic conditions, such as diabetes and cardiovascular diseases, all known to influence the ecosystem of the gut [11]. When it comes to shaping the profile of gut microbiota, diet is thought to be one of the main drivers behind this [7]. What we eat can promote the growth of varying bacterial strains that, in turn, alter fermentative metabolism and intestinal pH, which can then be responsible for the establishment of pathogenic bacteria [12]. Repeatedly, the composition of the diet and nutritional status have been found to be some of the most crucial modifiable factors controlling gut microbiota [13]. Now, there appears to be emerging evidence that what we drink could impact intestinal flora too.

As shown in Table 1, there are multiple compounds present in tea that could potentially interact with the gut microbiome. Tea polyphenols are absorbed by gut microbiota and considered to have an extended role in modern nutrition [14]. In particular, these are thought to have a ‘bi-directional’ relationship with gut flora by: (1) influencing gut microbiota composition (independently linked to health benefits) and (2) enabling gut microbiota to metabolise polyphenols and yield bioactive compounds (with potential clinical health benefits) [15]. 

In black tea, the major active polyphenols include the theaflavins and thearubigins and, in green tea, catechins are the prime polyphenolic compound which include epigallocatechin-3-gallate (EGCG; the most studied) along with epicatechin-3-gallate, epigallocatechin and epicatechin, gallocatechins and gallocatechin gallate [16]. The gut microbiota play a crucial role in the absorption of these compounds, with around 80% being absorbed and eventually excreted in urine [14]. Other recent work has also shown that tea can be a significant provider of ellagitannins, which appear to be converted by gut microbiota to urolithins that have been detected in human urine after tea consumption [17].

Increasingly, the gut microbiota has been identified as a key player involved in the aetiology of many intestinal and extra intestinal diseases [18]. It is coming to light that gut bacteria can break down the polyphenolic skeleton and perform reduction, decarboxylation, demethylation, and dihydroxylation reactions, which yield metabolites that can be absorbed in situ [19]. There are two dominant groups of beneficial bacteria in the adult gut—the *Bacteriodetes* and *Firmicutes*—with a growing body of evidence that proportions of *Bacteroidetes* are reduced in obese individuals, implying that obesity also has a microbial element [20]. Alongside this, there is science to suggest that gut microbiota metabolise polyphenols, which could counteract some of these effects [21]. 

Given these progressions, the present review aims to collate evidence from trials and mechanistic studies looking at tea and the gut microbiome. Such a review does not appear to have been undertaken previously.

## 2. Materials and Methods 

### 2.1. Study Selection

The Cochrane library and National Institutes of Health’s National Library of Medicine PubMed databases were used to identify relevant publications studying inter-relationships between tea components and the gut microbiome. 

Filters were applied and the inclusion criteria was defined as: (1) articles with an English abstract, (2) papers published in the last 10 years, (3) human trials that recruited free-living, healthy populations, and (4) papers studying tea, or tea components and gut health. Phase 1 comprised of a comprehensive search for human trials. Observational studies were not included as they are lower down in the hierarchy of scientific evidence and cause and effect relationships are more difficult to determine from these studies [24]. Similar key terms were used in Phase 2 but, this time, ‘animal and mechanistic studies’ published in the last five years were identified.

### 2.2. Search Strategy

The following search terms were applied to both Phases 1 and 2 of the review: Tea [tiab] AND micro* [tiab], intest* [tiab], colon* [tiab] or gut [tiab]. The Boolean term AND was used to focus the search, whilst the wild card asterisk (*) was used to find publications using varying terminologies such as: microbes, microbiota, microbiome or microflora. Studies focusing on vine tea, general colonic health, the oral microbiome or tea ingested via functional foods were excluded. Reference lists were also searched for further relevant publications. ED and TB identified and screened the articles which were compared against the inclusion/exclusion criteria. No disputes were apparent.

Phase 1 of the review followed the Preferred Reporting Items for Systematic Reviews and Meta-Analyses (PRISMA) statement [25]. The PRISMA flow algorithm used to depict the flow of information and map-out reasons for the exclusion of publications is shown in Figure 1 [25]. Once identified, all relevant trials had data extracted, including: Study (author, year, location), subjects (age, gender), study design, tea intervention (type, dosage) and main findings (Table 2). To establish quality scores for each human trial, the Jadad criterion was applied with scores of 1–5 being allocated (5 was indicative of higher quality trials; Table 2).

## 3. Results

The search identified 135 trials using the applied search terms. An additional two trials were found from reference lists (n = 137 total). Of these, four replica papers were removed, leaving 133 articles for abstract screening. A further 125 were then excluded; 101 were irrelevant, 14 were review papers, seven were irrelevant and focused on the oral microbiome or green tea mouth rinses and three used multi-component supplements. This left eight papers for further evaluation. Of these, a further two were excluded—one was a murine study and the other focused on absorption rather than the gut microbiome.

The algorithm of qualifying publications is shown in Figure 1. Of these, two studies were conducted in the United States, two in Asia (Japan and China), one in the Netherlands and one in Italy. Regarding animal and mechanistic studies, a total of 496 were identified, with 478 being excluded because they were irrelevant (*n* = 429), review papers (*n* = 46), or not published in the English language (*n* = 3), leaving 18 papers in the main review.

### 3.1. Evidence from Trials

Six trials have been conducted over the last ten years looking at tea compounds and their effects on the gut microbiome [26,27,28,29,30,31]. Of these, five focused on associations between green tea and the gut microbiome [26,27,28,30,31]. Three of these were beverage trials. One 24-hour feeding trial provided 400 mL daily of a ready-to-drink industrially made bottled beverage using Sri Lankan tea leaves (equivalent to about two cups)—a process based on a hot water infusion resembling traditional tea preparation [31]. The beverage supplied 400 µmol of flavan-3-ols with a calculated bioavailability of 39% and showed great variability in levels of excreted catechin metabolites (possibly due to different colonic microflora profiles) [31]. Unfortunately, whilst the green tea intervention provided –epicatechin, -epigallocatechin, -epigallocatechin-3-gallate, -epicatechin-3-gallate and gallic acid it also contained sugar, dextrose, lemon juice, ascorbic acid and flavor, which could have confounded results.

A 2-week intervention also supplied similar moderate amounts of green tea (400 mL daily). Green tea leaves were purchased from a Chinese local market and prepared using 400 mL deionised water boiled to 100 ℃ [27]. Following this, 3 g green tea leaves were dipped into the boiling water for 10 min and then removed—a process again aiming to simulate green tea preparation in daily life [27]. This level of ingestion led to significant alterations in gut microbiota composition, including an improved *Bifidobacterium* to *Enterobacteriacea* ratio [27]. A longer 10-day trial providing a higher daily intake of green tea—1000 mL (equivalent to about 4–5 cups) demonstrated prebiotic effects and also improved the colonic environment by increasing the proportion of *Bifidobacterium* species in faecal samples [30].

Two green tea trials used supplements or extracts. One 12-week randomised trial provided nine capsules daily, providing >1.35 g catechins, >0.56 g of epigallocatechin gallate and 1.80–1.97 g polyphenols, with no effects being observed on gut microbiota composition [28]. Although compliance checks were undertaken, it is plausible that not all nine capsules were taken daily which could have contributed to a lack of findings. A longer 12-month study providing capsules (2 daily) containing 0.84 g epigallocatechin gallate showed that microbial metabolism was involved in the formation of green tea polyphenol metabolites and inhibited the formation of aromatic amino acid metabolism, which, together, could have a favourable role on human health [26].

One trial conducted over a 30-hour period provided 12 healthy men with 2650 mg Brook Bond red label extract dissolved in 250 mL (1 mug) of hot water [29]. Results showed that levels of gut microbial catabolites varied considerably between individuals, which could have been attributed to different gut microbiota profiles [29]. Ongoing larger high-quality trials using tea in beverage form, higher levels of intakes and similar baseline body weights are now needed. 

Regarding quality, two of the identified trials were considered to be ‘high’ quality according to the Jadad scoring system [26,29]. The remainder were low–moderate quality. Future studies would benefit from rigorous randomisation methods with detailed assessments of subject compliance and withdrawal.

### 3.2. Evidence from Mechanistic Studies

A growing number of mechanistic studies have been undertaken in the last five years. Using the specified search criteria, 18 publications were identified. Within these, some studies used more than one tea form, with nine focusing on green tea [22,32,33,34,35,36,37,38,39] and three black tea [22,33,34] or their associated polyphenols. Two used Pu-er tea/extract [40,41], two Fuzhuan tea [42,43] and two oolong tea [22,44]. A range of methods were used, ranging from simulated gastrointestinal digestion, to murine models and microbiome–metabolome analysis.

Of those focusing on green tea and its polyphenols, three observed reductions in *Firmicutes* and improved levels of *Bacteroidetes*—changes that could help to prevent gut dysbiosis [34,38,39]. Several animal studies also found that tea ingestion (green, Fuzhuan, associated polyphenols) could help to ameliorate some of the unfavourable changes in microbial diversity bought about by high-fat diets and/or obesity [35,37,38]. Black tea exposure was found to have similar effects to green tea consumption using murine models—improving the diversity of microbiota and ratio of *Bacterioidetes* to *Firmicutes* [22,34].

Interestingly, emerging teas of interest, such as Pu-erh, Fuzhuan and Oolong, also brought about some favourable changes in gut profiles. For example, murine research showed that Pu-erh ingestion improved markers of metabolic syndrome possibly mediated via remodelling of gut microbiota [41]. Uzhuan brick tea ingestion helped to increase the diversity of microbiota in mice fed a high-fat diet [42]. Another study using pyrosequencing on rats found Fuzhuan tea consumption was linked to a threefold rise in *Lactobacillus spp* [43]. Improvements in *Bacteroidetes* levels and reductions in *Firmicutes* were observed in murine models ingesting oolong tea implying prebiotic effects [44].

Ongoing research is needed to better understand the mechanisms behind some of these study findings. In terms of potential theories, it has been found that the black tea theaflavin skeleton is fairly resistant to degradation by colonic bacteria with 67% recovery being identified and the generation of 21 phenolic and aromatic catabolites being reported [45]. Other work has found that tea catechins may inhibit pathogenic bacteria whilst stimulating the establishment of favourable bacteria [30,46,47]. It is possible that some of these catabolites could influence the profile of the gut microbiome. Another possibility is that tea ingestion could influence bile acid metabolism and affect bile acid receptor activation (FXR and/or TGR5) [48]. Additionally, tea compounds may act as bile acid receptor ligands influencing bile acid composition and, in turn, modulate microbiota composition and activate beneficial pathways [49]. Finally, it is also important to consider that the caffeine present in tea could also alter gut flora, although only a limited number of studies have considered this, with mixed results [50,51].

## 4. Discussion

A summary of results from this review are shown in Table 2 and Table 3. Most consistently, evidence for green tea appears to be linked to improved gut microbiota profiles, as seen in human and mechanistic studies. Regarding dosage and form, up to 1000 mL green tea daily (equivalent to 4–5 cups) has been associated with improved colonic bacterial profiles, including increased *Bifidobacteria* [30]. Emerging evidence for black, Pu-erh, Fuzhuan and oolong tea also looks promising, with similar mechanisms being reported—increases in *Bacteroidetes* and reductions in *Firmicute* populations [34,38,40,44]. 

Beyond the scope of this review, three prior trials have further looked into the effects of tea drinking and related polyphenols on intestinal and faecal microflora. An *in vivo* trial conducted on eight adults found that tea polyphenols (0.4 g/volunteer) given thrice daily over 4 weeks led to significant reductions in *Clostridium perfringens* and *Clostridium spp.* counts [47]. Similarly, a double-blind randomised trial found that drinking five servings of black tea daily inhibited some groups of faecal bacteria, but conclusive findings could not be drawn as microbial analysis was limited [55,56]. Other work in 2006 using faecal homogenates showed that tea phenolics significantly suppressed the growth of pathogenic bacteria, including *Clostridium perfringens, Clostridium difficile* and *Bacteroides spp.,* whilst favourable bacteria like *Lactobacillus sp.* were less severely affected [46]. These earlier studies add to the present body of evidence that tea drinking could help to induce favourable modulate gut microbiota profiles.

A number of studies have shown that tea drinking could be associated with weight loss [57,58,59] and it now seems plausible that the effects could, in part, be attributed to change at the gut microbiome level. For example, an *in vitro* study has shown that black tea, green tea and oolong tea extracts can all increase the growth of beneficial bacteria in the human intestine [60]. In an obese adult population, the ratio of *Firmicutes/Bacteroidetes* has been found to be higher than normal compared with lean weight people [61]. Together, these findings imply that tea and its polyphenol ingredients could have prebiotic activity, modulating the ratio of the types of bacteria in the gut which, in turn, could contribute to weight loss.

Overall, in the present review, the vast majority of studies showed positive modulatory effects. More research is now needed to understand ‘how’ teas modify gut microbiota along with levels of habitual consumption that would be required amongst healthy, normal-weight adults. The role of tea consumption in helping to alleviate symptoms of specific gastrointestinal disorders is also worthy of further exploration. A recent review concluded that polyphenol exposure using rodent models helped to inhibit colitis [62]. A pilot study involving patients with mild to moderate ulcerative colitis provided with polyphenols (up to 800 mg of (-)-epigallocatechin-3-gallate) found that the remission rate was 53% compared with 0% in the placebo, suggesting promise [63]. This work should be continued and further compared against placebo beverages, such as water.

In terms of strengths and limitations, some studies excluded subjects using antibiotics and others were stringent in asking participants not to drink any other form of tea, coffee or drinks providing polyphenols through the study duration, but further rigour is needed. Janssens et al. (2016), in particular, found that microbial diversity was significantly lower in overweight compared with normal-weight subjects, concluding that this too could act as a study confounder and required better consideration in further studies [28].

It is also important to consider that large inter and intra-individual variations exist in the human faecal microflora composition [56], which is one reason why ongoing, larger and adequately powered trials are needed. Gender is another factor that can influence gut microbiota, potentially acting as a confounder in studies, with a need to further investigate how gut flora varies between genders [64]. There was also variability in sample sizes and study durations which could account for some of the heterogeneity in results. 

Larger, longer randomised and blinded human studies are necessary to better understand the effect of teas on the human intestinal and faecal microbiota. It is important that there continues to be further uniformity in tea forms used in future studies. For example, some used supplements, a bolus or beverages. From a practical viewpoint, interventions are required that can be translated in real-life terms and applied from a public health stance. This is important, as tea drinking could represent an attractive adjunctive lifestyle tool for helping to support gut health.

Overall, given the totality of the evidence, it seems logical to suggest that tea drinking, but especially green tea, could help to improve the profile of gut microbiota and even exhibit prebiotic effects. In particular, provisional evidence from animal studies suggests that these benefits may be more prominent amongst those with high-fat diets, obesity or insulin resistance but this is yet to be reconfirmed by human trials [37,38,54]. In the meantime, as evidence builds, tea remains to be an important source of polyphenols, which have an extensive role in promoting human health [16].

## 5. Conclusions

There has been a growing body of evidence evaluating inter-relationships between tea compounds and the gut microbiome. Presently, most of this is from mechanistic studies with a number of human trials emerging. Initial evidence suggests that teas could benefit gut health, particularly amongst those with dysbiosis (triggered by obesity or high-fat diets), mainly by mediating gut microbiota profiles. At the moment, evidence appears to be strongest for green tea and emerging for black, oolong, Pu-erh and Fuzhuan teas. Longer, follow-up trials are now needed to better understand inter-relationships between tea compounds and the gut microbiome.

## Figures and Tables

**Figure 1 nutrients-11-02364-f001:**
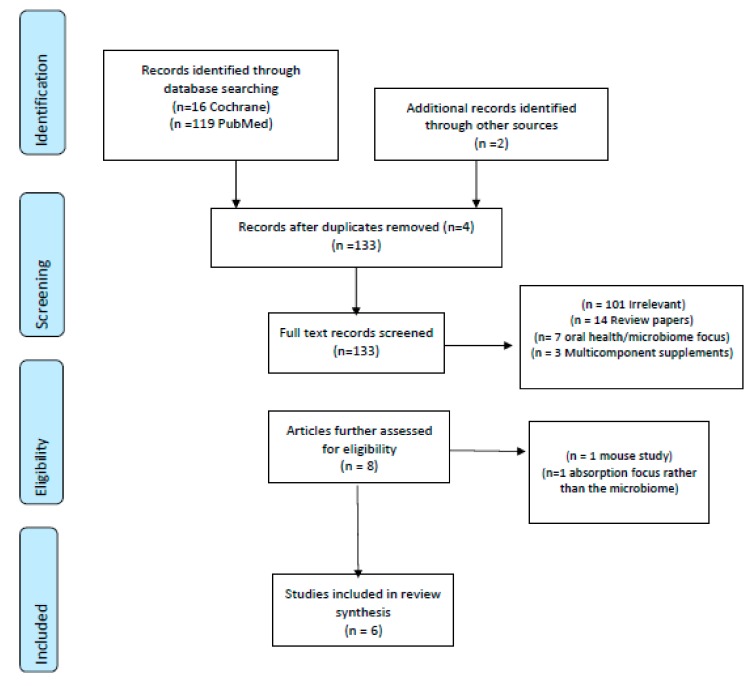
PRISMA algorithm used to identify trials [25].

**Table 1 nutrients-11-02364-t001:** Tea compounds and the gut microbiome.

Tea Compounds	Reference
**Ellagitannins**—Tea has been found to be a significant contributor of dietary ellagitannins, which the gut microbiota metabolites use to produce urolithins.	Yang et al. (2019) [17]
**Oligomeric, oxidized black tea phenolic (BTP) and monomeric green tea catechin (GTC)**—GTC gives a higher yield of bioactive phenolic metabolites upon colonic fermentation than BTP.	Liu et al. (2016) [22]
**Tea polyphenols**—The major classes are catechins, including epicatechin, epigallocatechin, epicatechin-3-gallate, and epigallocatechin-3-gallate. Flavanols, such as quercetin, kaempferol, myricetin and their glycosides, are also found which could interact with gut microbiota.	Etxeberria et al. (2013) [8]
**Hippuric acid**—Ingestion of green and black tea majorly increases the excretion of hippuric acid into urine, though less is known about microbial degradation.	Mulder et al. (2005) [23]

**Table 2 nutrients-11-02364-t002:** Phase 1: Tea compounds and the gut microbiome: Evidence from human trials.

Study (Author, Year, Location, Reference Number)	Subjects (age, gender)	Study design	Tea Intervention (type)	Tea Intervention (dosage)	Main Findings	JADAD Score
Zhou et al. (2019) United States [26]	Postmenopausal F.	12-month intervention.	GTP supplement.	GT catechin extract containing 843.0 ± 44.0 mg/day epigallocatechin gallate or placebo capsules for 1 year.	Microbial metabolism of GTP and aromatic amino acids appear to play a role in the health effects of GT consumption in humans.	5
Yuan, et al. (2018) China [27]	*n* = 12 healthy subjects, M and F, 27–46 years.	2-week intervention.	GT beverage.	400 mL green tea daily. One-week washout and 2-week intervention.	An irreversible, increased *Firmicutes* to *Bacteroidetes* ratio was observed along with a reduction of bacterial LPS synthesis in faeces after GTL ingestion.	0
Janssens et al. (2016) United States [28]	*n* = 58 Caucasian, M and F, 18–50 years.	12-week randomised, single blind, placebo-controlled design.	GT capsules.	Capsules with GT extract (containing >0.06 g Epigallocatechin-3-gallate and 0.03–0.05 g caffeine per capsule). Nine capsules were taken daily.	Significant effects on composition of the gut microbiota were not observed although a reduced bacterial alpha diversity in overweight vs. normal-weight subjects was seen (*p* = 0.002).	2
Van Duynhoven et al. (2014) Netherlands [29]	*n* = 12 healthy men.	30-hour randomised, open, placebo-controlled, crossover study.	Single bolus of BTE.	2650 mg of Brook Bond red label extract, dissolved in 250 mL of hot water.	Inter-individual variation in response was greater for gut microbial catabolites than for directly absorbed BTPs. Rapid and sustained circulation of conjugated catabolites suggests these may be relevant to BTE health benefits.	4
Jin et al. (2012) Japan [30]	*n* = 10 non habitual green tea drinkers, M and F, 33–70 years.	17-day trial (intervention for 10 days).	GT beverage.	1000 mL of GT daily. Drank GT instead of water for 10 days.	There was an overall tendency for the proportion of *Bifidobacteria* to increase due to GT ingestion. GT consumption may act as a prebiotic and improve the colon environment by increasing the proportion of the *Bifidobacterium* species.	1
Del Rio et al. (2010) Italy [31]	*n* = 20 healthy subjects.	24-hour feeding trial.	GT beverage.	400 mL of a RTD GT containing approximately 400 μmol of flavan-3-ols.	Colonic microflora-derived polyhydroxyphenyl-γ-valerolactones were the main urinary catabolites, averaging 10 times greater concentration than flavan-3-ol conjugates.	0

Key: BTE, Black Tea Extract; BTP, Black Tea Polyphenols; F, Female; GT, GTE, Green Tea Extract; Green Tea; GTL, Green Tea Liquid; GTP, Green Tea Polyphenol; LPS, lipopolysaccharide; M, male; RTD, Ready-To-Drink. Source: Jadad, A.R. et al. *Control Clin Trials*, 1996. 17(1): pp. 1–12.

**Table 3 nutrients-11-02364-t003:** Phase 2: Tea compounds and the gut microbiome: Evidence from mechanistic studies.

Study (Author, Year, Reference Number)	Study Design	Tea Intervention (type)	Main Findings
Lu et al. (2019) [52]	Obese murine study.	Ripened Pu-erh tea extract.	Ripened Pu-erh tea extract could potentially prevent obesity through rebalancing the gut microbiota.
Xia et al. (2019) [40]	Metagenomic/meta-proteomic using obese rats.	Aqueous raw and ripe Pu-erh tea extracts.	Raw and ripe Pu-erh teas, administration at two doses significantly increased microbial diversity and changed the composition of cecal microbiota by increasing *Firmicutes* and decreasing *Bacteroidetes.*
Zhang et al. (2019a) [32]	Animal and human *in vitro* studies.	(-)-epigallocatechin-3-gallate and green tea.	Microbiota facilitates the formation of the aminated metabolite of green tea polyphenol (-)-epigallocatechin-3-gallate which trap reactive endogenous metabolites.
Zhang et al. (2019b) [53]	Diabetic murine study.	Corn-starch tea.	Corn-starch‒tea diet resulted in reduced blood glucose, increased levels of *Coriobacteriaceae*, *Lactobacillaceae*, *Prevotellaceae* and *Bifidobacteriaceae,* and decreased *Bacteroidaceae, Ruminococcaceae, Helicobacteraceae* and *Enterobacteriaceae.*
Zhou et al. (2019) [26]	Human study.	Green tea polyphenols.	GTP may have anti-obesity actions namely via changes in gut-microbiota metabolism.
Annunziata et al. (2018) [33]	Simulated GI digestion.	Tea polyphenols from green, white and black tea.	Gut microbiota appear to metabolise polyphenols generating metabolites with a greater antioxidant activity.
Chen et al. (2018a) [54]	Normal and obese rats.	Tea polyphenols.	A high-fat high sugar diet appeared to influence the excretion of tea catechins, leading to insufficient metabolism of catechins by the gut microflora.
Chen et al. (2018b) [42]	Murine study.	Fuzhuan brick tea polysaccharides.	Increased the phylogenetic diversity of high-fat diet-induced microbiota. Could help prevent modulation of gut microbiota.
Cheng et al. (2018) [44]	Murine study.	Oolong tea polyphenols.	A large increase in *Bacteroidetes* with a decrease in *Firmicutes* was observed having a positive modulatory and prebiotic effect.
Cheng et al. (2017) [39]	Mice model.	(-)-Epigallocatechin 3-O-(3-O-methyl) gallate.	A large increase in *Bacteroidetes* with concomitant decrease of *Firmicutes* was observed after the administration of EGCG3 for 8 weeks. Could help to prevent gut dysbiosis.
Henning et al. (2018) [34]	Murine study.	Green and black tea polyphenols.	GTP and BTPs decreased cecum *Firmicutes* and increased *Bacteroidetes.*
Wang et al. (2018) [35]	Human flora-associated C57BL/6J mice model.	Green tea polyphenols.	A high-fat diet significantly impacted gut microbiota composition and lipid metabolism which was ameliorated by tea polyphenols.
Gao et al. (2017) [41]	Murine study.	Pu-erh tea.	Post fermented pu-erh tea providing polyphenols and caffeine improved diet-induced metabolic syndrome which was attributed to remodelling of the gut microbiota.
Jung et al. (2017) [36]	Murine microbiome-metabolome analysis.	Green tea supplementation.	Green tea supplementation improved the microbial community diversity by altering states of various endogenous metabolites in mice groups subjected to UVB-exposure.
Foster et al. (2016) [43]	Pyrosequencing using rats.	Fuzhuan tea.	Fuzhuan tea altered intestinal function and was associated with a threefold increase in two *Lactobacillus spp.*
Liu et al. (2016) [22]	Obese C57BL/6J mice.	Green, oolong and black tea.	Tea infusion consumption substantially increased diversity and altered the structure of gut microbiota.
Wang et al. (2016) [37]	C57BL/6J Human Flora-Associated mice.	Green tea polyphenols.	High-fat diet was associated with a significant reduction in microbial diversity which was alleviated by tea polyphenol ingestion.
Seo et al. (2015) [38]	Murine study.	Fermented green tea extract.	Fermented green tea restored the changes in gut microbiota composition (e.g., the *Firmicutes/Bacteroidetes* and *Bacteroides/Prevotella* ratios) closely linked to development of obesity and insulin resistance, induced by high-fat diets.

Key: BTP, Black Tea Polyphenols; GI, gastrointestinal; GTP, Green Tea Polyphenols; TF, theaflavin; TF3G, theaflavin-3-gallate; TF3’G, theaflavin-3’-gallate.

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
