# Peer review of "Tea Compounds and the Gut Microbiome: Findings from Trials and Mechanistic Studies"

_nutrients, 2019, doi:10.3390/nu11102364_

Round 1

Reviewer 1 Report

This is a novel study on an interesting topic, and the results could be very relevant to many people from tea-drinking cultures around the world. 

The presentation of the results is sound, and the PRISMA guideline has been used to identify trials and create the flowchart. It would have been nice if the review had been registered in PROSPERO during the planning phase, as this helps reduce bias. 

In most part the study is of good quality, however I do have some concerns, in particular about the search strategy: 

Why was only one database (Pubmed) included? A systematic review should not compromise sensitivity by only including one database. Pubmed is a free database that uses the database MEDLINE, and it does not capture all the citations found in the subscription-based databases such as Embase (Elsevier). In addition the Cochrane library should be added.  Articles in non-English language should be added to reduce publication bias, at least if they have an English abstract.  Inclusion and exclusion criteria need to be stated more clearly.  The whole articles (fulltext) should be automatically searched for keywords, not just the title and the abstract. Then, a manual search should be done using the title and the abstract, before deciding on the final papers to review.  How many researchers performed the review? Ideally, two independent researchers should select the articles and review them, and a third one should resolve any disputes.  Why were not observational studies included? 

Additional comments: 

The introduction could benefit from som re-structuring, and more background should be added about the gut-microbiota (what other factors could affect it, and which factors could confound the results?).  The description of the phase 2 (line 82-84) is not clear The limitations of the included studies should be discussed in further depth. What about sample size and heterogeneity of the studies?  The conclusion made is too strong based on the evidence provided. We need longer, follow-up trials to conclude about whether tea compounds are beneficial to the gut microbiome. As of now, we are not even certain which microbes are more beneficial than other to human health. 

Author Response

Many thanks. 

This feedback has been included and highlighted in yellow.  An additional Cochrane database search was undertaken.  Unfortunately Embase subscription was estimated to be £12,000 so this was too costly to undertake.

All the remaining feedback has been applied.

Reviewer 2 Report

The authors performed a systematic review of human and animal studies related to tea consumption and its effects on the gut microbiota. It is one of the first, if not the first to review specifically this topic. It is well written and the search strategy and study selection well described. The tables summarizing findings were appreciated.

Comments:

This reviewer noted that some animal studies were omitted (see below). The findings from these studies add to mechanism. Were they overlooked or did they not pass the screening criteria. July 1st was a cut-off for the search, but the second and third papers were published in May and June and the first paper was in published electronically in April. If they would have passed the inclusion criteria, then the authors should consider adding them and double check that additional studies have not been overlooked.

Ushiroda C, Naito Y, Takagi T, Uchiyama K, Mizushima K, Higashimura Y, Yasukawa Z, Okubo T, Inoue R, Honda A, Matsuzaki Y, Itoh Y. Green tea polyphenol (epigallocatechin-3-gallate) improves gut dysbiosis and serum bile acids dysregulation in high-fat diet-fed mice. J Clin Biochem Nutr. 2019 Jul;65(1):34-46. doi: 10.3164/jcbn.18-116. Epub 2019 Apr 6. PubMed PMID:31379412; PubMed Central PMCID: PMC6667385.

Zhang HH, Liu J, Lv YJ, Jiang YL, Pan JX, Zhu YJ, Huang MG, Zhang SK. Changes in Intestinal Microbiota of Type 2 Diabetes in Mice in Response to Dietary Supplementation With Instant Tea or Matcha. Can J Diabetes. 2019 May 8. pii: S1499-2671(18)30865-7. doi: 10.1016/j.jcjd.2019.04.021. [Epub ahead of print] PubMed PMID: 31378691.

Lu X, Liu J, Zhang N, Fu Y, Zhang Z, Li Y, Wang W, Li Y, Shen P, Cao Y. Ripened Pu-erh Tea Extract Protects Mice from Obesity by Modulating Gut Microbiota Composition. J Agric Food Chem. 2019 Jun 26;67(25):6978-6994. doi: 10.1021/acs.jafc.8b04909. Epub 2019 Jun 11. PubMed PMID: 31070363.

While the authors mention mechanism, there is a lack of discussion about what mechanisms may be involved other than changes in microbiota composition. Based on the studies reviewed, are potential mechanisms discussed? For example, could altered bile acid metabolism affect bile acid receptor activation (FXR and/or TGR5)? The tea compounds may also act as bile acid receptor ligands (see publications below) and this could affect bile acid composition that may influence microbiota composition and activate beneficial pathways. A bit more thought and discussion along these lines would benefit the manuscript.

Sheng L, Jena PK, Liu HX, Hu Y, Nagar N, Bronner DN, Settles ML, Bäumler AJ, Wan YY. Obesity treatment by epigallocatechin-3-gallate-regulated bile acid signaling and its enriched Akkermansia muciniphila. FASEB J. 2018 Jun8:fj201800370R. doi: 10.1096/fj.201800370R. [Epub ahead of print] PubMed PMID:29882708; PubMed Central PMCID: PMC6219838.

Li G, Lin W, Araya JJ, Chen T, Timmermann BN, Guo GL. A tea catechin, epigallocatechin-3-gallate, is a unique modulator of the farnesoid X receptor. Toxicol Appl Pharmacol. 2012 Jan 15;258(2):268-74. doi: 10.1016/j.taap.2011.11.006. Epub 2011 Dec 4. PubMed PMID: 22178739; PubMed Central PMCID: PMC3259191.

Author Response

Extra information regarding mechanisms has been added. 

The suggested papers have been added though one (Ushiroda did not meet the inclusion criteria so was not included - it did not mention the microbiome specifically).  

Round 2

Reviewer 1 Report

Some of the new sentences in yellow need revising, they are lacking commas and periods, and are "hanging".